# Heterogeneous associations of polygenic indices of 35 traits with mortality: a register-linked population-based follow-up study

**Hannu Lahtinen[1,2]\*, Jaakko Kaprio[3], Andrea Ganna[4], Kaarina Korhonen[1,2], Stefano Lombardi[4,5,6], Karri Silventoinen[1], Pekka Martikainen[1,2]**

[1]Helsinki Institute for Demography and Population Health, University of Helsinki, Helsinki, Finland; [2]Max Planck – University of Helsinki Center for Social Inequalities in Population Health, Helsinki, Finland; [3]Institute for Molecular Medicine FIMM, University of Helsinki, Helsinki, Finland; [4]FIMM, University of Helsinki, Helsinki, Finland; [5]VATT Institute for Economic Research, Helsinki, Finland; [6]Institute of Labor Economics (IZA), Bonn, Germany

## eLife Assessment

This **important** study reports **convincing** evidence of associations between 35 polygenic indices (PGIs) for social, behavioural, and psychological traits, as well as other health conditions (e.g., BMI) and all-cause mortality, based on data from Finnish population-based surveys and a twin cohort linked to administrative registers. PGIs for education, depression, alcohol use, smoking, BMI, and self-rated health showed the strongest associations with all-cause mortality, in the order of ~10% increment in risk per PGI standard deviation. Effect sizes from twin-difference analyses tended to be slightly larger than those from population cohorts, a pattern opposite that generally observed when testing PGI associations with their target phenotypes, and supporting the robustness of findings to confounding by population stratification.

\*For correspondence:
hannu.lahtinen@helsinki.fi

**Competing interest:** The authors declare that no competing interests exist.

## Abstract

**Background:** Polygenic indices (PGIs) of various traits abound, but knowledge remains limited on how they predict wide-ranging health indicators, including the risk of death. We investigated the associations between mortality and 35 different PGIs related to social, psychological, and behavioural traits, and typically non-fatal health conditions.

**Methods:** Data consist of Finnish adults from population-representative genetically informed epidemiological surveys (FINRISK 1992–2012, Health 2000/2011, FinHealth 2017), linked to administrative registers (N: 40,097 individuals, 5948 deaths). Within-sibship analysis was complemented with dizygotic twins from Finnish twin study cohorts (N: 10,174 individuals, 2116 deaths). We estimated Cox proportional hazards models with mortality follow-up 1995–2019.

**Results:** PGIs most strongly predictive of all-cause mortality were ever smoking (hazard ratio [HR]=1.12, 95% confidence interval [95% CI] 1.09; 1.14 per one standard deviation larger PGI), self-rated health (HR = 0.90, 95% CI 0.88; 0.93), body mass index (HR = 1.10, 95% CI 1.07; 1.12), educational attainment (HR = 0.91, 95% CI 0.89; 0.94), depressive symptoms (HR = 1.07, 95% CI 1.04; 1.10), and alcohol drinks per week (HR = 1.06, 95% CI 1.04; 1.09). Within-sibship estimates were approximately consistent with the population analysis. The investigated PGIs were typically more predictive for external than for natural causes of death. PGIs were more strongly associated

with death occurring at younger ages, while among those who survived to age 80, the PGI–mortality associations were negligible.

**Conclusions:** PGIs related to the best-established mortality risk phenotypes had the strongest associations with mortality. They offer moderate additional prediction even when mutually adjusting with their phenotype.

**Funding:** HL was supported by the European Research Council [grant #101019329] as well as the Max Planck – University of Helsinki Center for Social Inequalities in Population Health. SL gratefully acknowledges funding from the Research Council of Finland (# 350399). PM was supported by the European Research Council under the European Union's Horizon 2020 research and innovation programme (#101019329), the Strategic Research Council (SRC) within the Research Council of Finland grants for ACElife (#352543-352572) and LIFECON (#345219), the Research Council of Finland profiling grant for SWAN (#136528219) and FooDrug (# 136528212), and grants to the Max Planck – University of Helsinki Centre for Social Inequalities in Population Health from the Jane and Aatos Erkko Foundation (#210046), the Max Planck Society (# 5714240218), University of Helsinki (#77204227), and Cities of Helsinki, Vantaa and Espoo (#4706914). The study does not necessarily reflect the Commission's views and in no way anticipates the Commission's future policy in this area. The funders had no role in the study design, data collection and analysis, decision to publish, or preparation of the manuscript.

## Introduction

The genome-wide association study (GWAS) literature has identified a vast number of polygenic indices (PGIs) on almost every widely-measured human phenotype (*Burt, 2024*; *Becker et al., 2021*; *Choi et al., 2020*). One motivation for this endeavour has been to construct tools that may help clinical practice in disease risk prediction (*Torkamani et al., 2018*; *Lambert et al., 2019*; *Lewis and Vassos, 2020*). In addition, PGIs can have possible utility for health risk assessment in a more indirect manner, since they may have downstream importance in predicting health and functioning more widely beyond their immediate phenotypes, such as regarding their ability to predict all-cause mortality. Previous studies have sought genetic variants and their composites to explain mortality (*Ganna et al., 2013*) or closely related concepts such as (disease-free) life span (*Jukarainen et al., 2022*; *Sakaue et al., 2020*) and biological ageing (*Argentieri et al., 2025*). Some previous studies have also assessed associations between PGIs of different traits and mortality (*Argentieri et al., 2025*; *Karlsson Linnér and Koellinger, 2022*; *Meisner et al., 2020*). However, the existing literature covers mostly disease- or biomarker-related PGIs, and our knowledge is still limited on to what extent PGIs for social, psychological, and behavioural phenotypes or PGIs for typically non-fatal health conditions can help in mortality prediction.

Additionally, the state-of-the-art knowledge is scarce regarding to what extent the PGI–mortality associations stem from direct genetic effects. When the interest lies beyond merely predictive uses, the research has increasingly shown the limitations of PGIs as black-box predictors, which may include – in addition to the usually desired direct genetic signals – population-related phenomena due to geographical stratification of ancestries, dynastic or shared environmental effects in families and kins, as well as assortative mating. Within-sibship analysis designs may alleviate such limitations, taking advantage of the fact that the genetic differences between siblings originate from the random segregation at meiosis (*Burt, 2024*; *Howe et al., 2022*).

Another area where the knowledge on the PGI–mortality relationship is still relatively limited includes heterogenous associations with socio-demographic factors, different contributions to causes of death, as well as how the associations interplay with corresponding phenotypes. The mortality risks between individuals differ substantially by their sex, age, and education (*Hoffmann et al., 2019*; *Rogers et al., 2010*), warranting an assessment of the potential heterogeneous effects regarding them. It is also possible that social, psychological, and behaviour-related PGIs may matter disproportionately more for certain causes of death, including accidents, suicides, and violent and alcohol-related deaths. Such 'external' mortality is conceptually closely connected to risky behaviour and substance use. A relevant question for their practical utility is also whether PGIs can bring additional information to predicting the risk of death in cases when information of the measured phenotype is available.

In this study, we address these gaps in knowledge by assessing the association between 35 different PGIs – mostly related to social, psychological, and behavioural traits or typically non-fatal health conditions – and mortality using a population-representative sample of over 40,000 Finnish individuals with up to 25 years of register-based mortality follow-up. We also assess the extent of potential population stratification and related biases by within-sibship analysis of over 10,000 siblings. Furthermore, we examine potential heterogeneous PGI–mortality associations by sex and education, as well as for mortality occurring at different ages and separately for external and natural causes of death. Finally, we compare six PGIs that show the strongest associations with mortality (PGIs of ever smoking, body mass index [BMI], depressive symptoms, alcoholic drinks per week, educational attainment and self-rated health) and their phenotypes when mutually adjusting for each other.

## Methods
### Study population
The main ('population') analysis sample consists of genetically informed population surveys FINRISK rounds 1992, 1997, 2002, 2007, and 2012, as well as Health 2000/2011 and FinHealth 2017 (*Borodulin et al., 2018*; *Borodulin and Sääksjärvi, 2019*; *Lundqvist and Mäki-Opas, 2016*). The response rates of these data collections varied between 65 and 93%. The genetic data followed the quality control and imputation procedures described in *Pärn et al., 2018a*; *Pärn et al., 2018b*. Among the initial pooled sample, 88% had genotyped data available after the quality-control procedures. These data were linked to administrative registers using pseudonymised personal identity codes, including socio-demographic and mortality information maintained by Statistics Finland. In addition to this population sample, we used a within-sibship analysis sample to assess the extent of direct and indirect genetic associations captured by the PGIs, as discussed in the introduction. Individuals in the sibling data were mainly dizygotic twins from Finnish twin cohorts: 'old cohort' (born before 1958), FinnTwin16 (born 1974–1979), and FinnTwin12 (1983–1987; *Kaprio, 2013*). These sibling data were complemented with individuals identified from the population sample as likely full siblings based on their genetic similarity (0.35<Identity by descent <0.80) and having an age difference of less than 18 years. They were excluded from the main population analysis to achieve non-overlapping samples.

The individuals were followed from (whichever was latest) (1) January of 1995, (2) July of the data collection year, or (3) the month the respondent turned 25 years. The mortality follow-up ended (whichever was earliest) at the end of 2019, or at the date of death. The analytic sample size was 40,097 individuals (564,885 person-years of follow-up) and 5948 deaths in the population analysis. The within-sibship analysis included 10,174 individuals (200,683 person-years of follow-up) in 5071 sibships and 2116 deaths.

### Variables
The outcome was death that occurred during the follow-up period. External (accidents, suicides, violence, and alcohol-related causes of death; International Classification of Diseases 10th revision codes: F10, G312, G4051, G621, G721, I426, K292, K70, K852, K860, O354, P043, Q860, V01–Y89; 587 deaths) and natural causes of death (other codes; 5349 deaths) were identified from the national cause of death register collected by Statistics Finland. Twelve individuals had an unknown cause of death and were excluded from the cause-specific analysis.

As the independent variables of main interest, we used 35 different PGIs in the Polygenic Index repository by *Becker et al., 2021*, which were mainly based on GWASes using UK Biobank and 23andMe, Inc data samples, but also other data collections. They were tailored for the Finnish data, that is excluding overlapping individuals between the original GWAS and our analysis and performing linkage-disequilibrium adjustment. We used every single-trait PGI defined in the repository (except for subjective well-being, for which we were unable to obtain a meta-analysis version that excluded the overlapping samples). By limiting the researchers' freedom in selecting the measures, this conservative strategy should increase the validity of our estimates, particularly with regard to multiple-testing adjusted p-values. The PGIs are described in *Supplementary file 1A* (see also *Becker et al., 2021*; *PGI Repository, 2025*). The PGIs were standardised to have mean 0 and standard deviation (SD) 1.

We also measured corresponding phenotypes for the six PGIs with the strongest association with mortality: smoking (never/quit at least six months ago/current), BMI (kg/m$^2$), depression (number of

indicators 0–3), alcohol intake (grams of ethanol per week), education (expected years to complete the highest attained degree), and self-rated health (1–5). *Supplementary file 1B* presents more information on the measurement of the phenotypes and their distributions. For parsimony and comparability to the PGIs, these phenotypes were also standardised to SD units in the main analysis, whereas analyses of categorically measured phenotypes are presented in the supplement. After excluding individuals with any missing information, analysis that included phenotypes had 37,548 individuals and 5407 deaths. *Supplementary file 1C* presents correlations between PGIs and studied phenotypes.

## Modelling

We estimated Cox proportional hazards regressions predicting mortality by each PGI. We used age as the time scale as recommended in *Thiébaut and Bénichou, 2004*. All the models were adjusted for indicators for the data collection baseline year, sex, and the ten first principal components of the full (pruned) single nucleotide polymorphism (SNP) matrix. The models were first estimated for the whole population sample. We compared these models to the corresponding within-sibship models, using the sibship identifier as the strata variable. This method employs a sibship-specific (instead of a whole-sample-wide baseline hazard in the population models) baseline hazard and corresponds to a fixed-effects model in some other regression frameworks (e.g. linear model with sibship-specific intercepts; *Allison, 2009*; *Rabe-Hesketh and Skrondal, 2012*).

Next, we assessed heterogeneous associations by estimating the corresponding models among men and women separately, as well as in three education groups. We also investigated possible age-related heterogeneous patterns, fitting the corresponding model in three age-specific mortality follow-up periods (25–64 years, 65–79 years, 80+ years). We also conducted an additional analysis by separating the outcome into external and natural causes of death.

Finally, we analysed the six PGIs with the strongest association to mortality in more detail. Here, we also measured the corresponding phenotypes and fitted four types of models: Model 1 was adjusted for controls and each PGI/phenotype separately. Model 2 jointly adjusted for corresponding PGI and phenotype. Model 3 adjusted for all PGIs or phenotypes (but not both simultaneously) and Model 4 adjusted for all six PGIs and phenotypes simultaneously. We also carried out an analysis stratified by whether the study participants did or did not have the phenotypic risk factor.

The proportional hazards assumption of Cox models was evaluated with Schoenfeld residuals (see *Supplementary file 1D*; *Allison, 2014*). Residual correlations of PGIs were no more than 0.050 of their absolute value. The correlations with investigated phenotypes were higher, however, particularly for (continuous) BMI (–0.099).

Multiple-testing adjustments of p-values were conducted with the Benjamini–Hochberg method (*Benjamini and Hochberg, 1995*) for 35 multiple tests.

The software used for producing genetic variables was PLINK versions 1.9 and 2.0. The statistical analysis was conducted with Stata versions 16 and 18. The code used for data preparation and analysis can be found at https://github.com/halahti/eLife26/ (copy archived at *Lahtinen, 2026*).

## Results

*Figure 1* displays the associations between 35 PGIs and all-cause mortality for the population analysis and within-sibship analysis samples. In the population analysis, the PGIs that showed the strongest associations with mortality were ever smoking (hazard ratio [HR] per 1 SD larger PGI = 1.12, 95% confidence interval [95% CI] 1.09; 1.15), self-rated health (HR = 0.90, 95% CI 0.88; 0.93), BMI (HR = 1.10, 95% CI 1.07; 1.13), educational attainment (HR = 0.91, 95% CI 0.89; 0.94), depressive symptoms (HR = 1.07, 95% CI 1.04; 1.10), and drinks per week (HR = 1.06, 95% CI 1.04; 1.09). Most of the studied PGIs had negligible associations with mortality, as 18 PGIs had HRs between 0.98 and 1.02.

Although CIs overlapped with regards to every individual PGI except extraversion, on average, within-sibship analysis had marginally larger associations than the population analysis (inverse-variance-weighted mean absolute log HR was 0.023, 95% CI 0.005; 0.042 larger in within-sibship than population analyses). In within-sibship analysis, the PGI of BMI had the strongest association with mortality (HR = 1.22, 95% CI 1.10; 1.36).

*Figure 2* presents PGI–mortality associations by sex, education, age, and cause of death. Overall, men had slightly stronger PGI–mortality associations (Panel A; inverse-variance-weighted mean

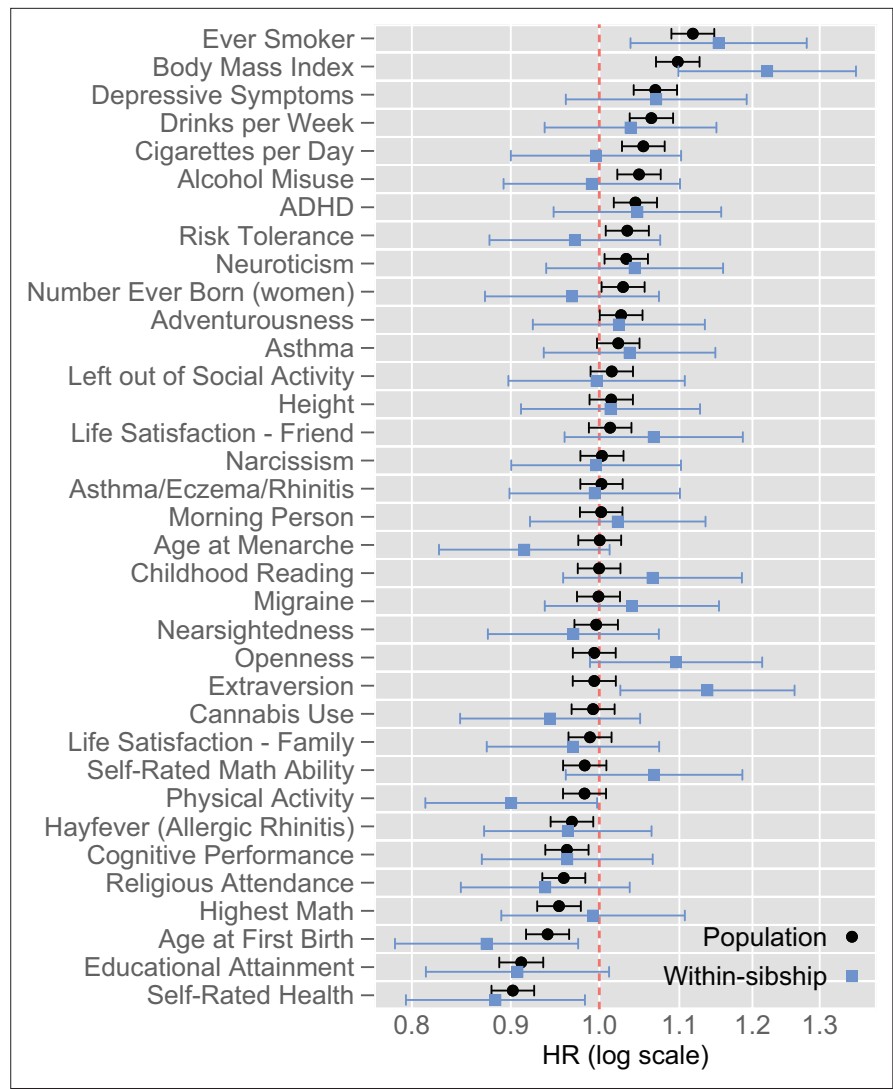

**Figure 1.** Hazard ratios of polygenic indices for all-cause mortality. Population (N=40,097 individuals) and within-sibship (N=10,174 individuals in 5071 sibships) estimates. Estimates from Cox proportional hazards models adjusted for indicators for the baseline year, sex and 10 first principal components of the genome. Capped bars are 95% confidence intervals. For a table of corresponding estimates, see **Supplementary file 1E**. Abbreviations: HR = Hazard ratio; ADHD = Attention deficit hyperactivity disorder.

absolute log HR was 0.013, 95% CI 0.005; 0.022 larger among men). The largest sex differences were observed for the PGI for educational attainment (HR = 0.89, 95% CI 0.86; 0.92 among men; HR = 0.95, 95% CI 0.91; 0.99 among women; p=0.010 for difference, p=0.21 after multiple-testing adjustment with Benjamini–Hochberg method for 35 joint tests) and PGI for attention deficit hyperactivity disorder (ADHD; HR = 1.08, 95% CI 1.04; 1.11 among men; HR = 1.01, 95% CI 0.97; 1.05 among women; p=0.012 for difference, p=0.21 after multiple-testing adjustment). The HRs in Panel B indicate no evidence for substantial heterogeneous associations by education level. Panel C analyses mortality in three age-specific follow-up periods. The PGIs were more predictive of death in younger age groups, although the difference between the 25–64 and 65–79 age groups was small, except for the PGI of ADHD (HR = 1.14, 95% CI 1.08; 1.21 for 25–64 year-olds; HR = 1.04, 95% CI 1.00; 1.08 for 65–79 year-olds; p=0.008 for difference, p=0.27 after multiple-testing adjustment). PGIs predicted death only negligibly among those aged 80+, and the largest differences between the age groups 25–64 and 80+were for PGIs of self-rated health (HR = 0.87, 95% CI 0.82; 0.93 for 25–64 year-olds, HR = 1.00, 95% CI 0.94; 1.04 for 80+ year olds, p=2*10$^{-4}$ for difference, p=0.006 after multiple-testing adjustment), ADHD (HR = 1.14, 95% CI 1.08; 1.21 for 25–64 year-olds, HR = 0.99, 95% CI 0.95; 1.03

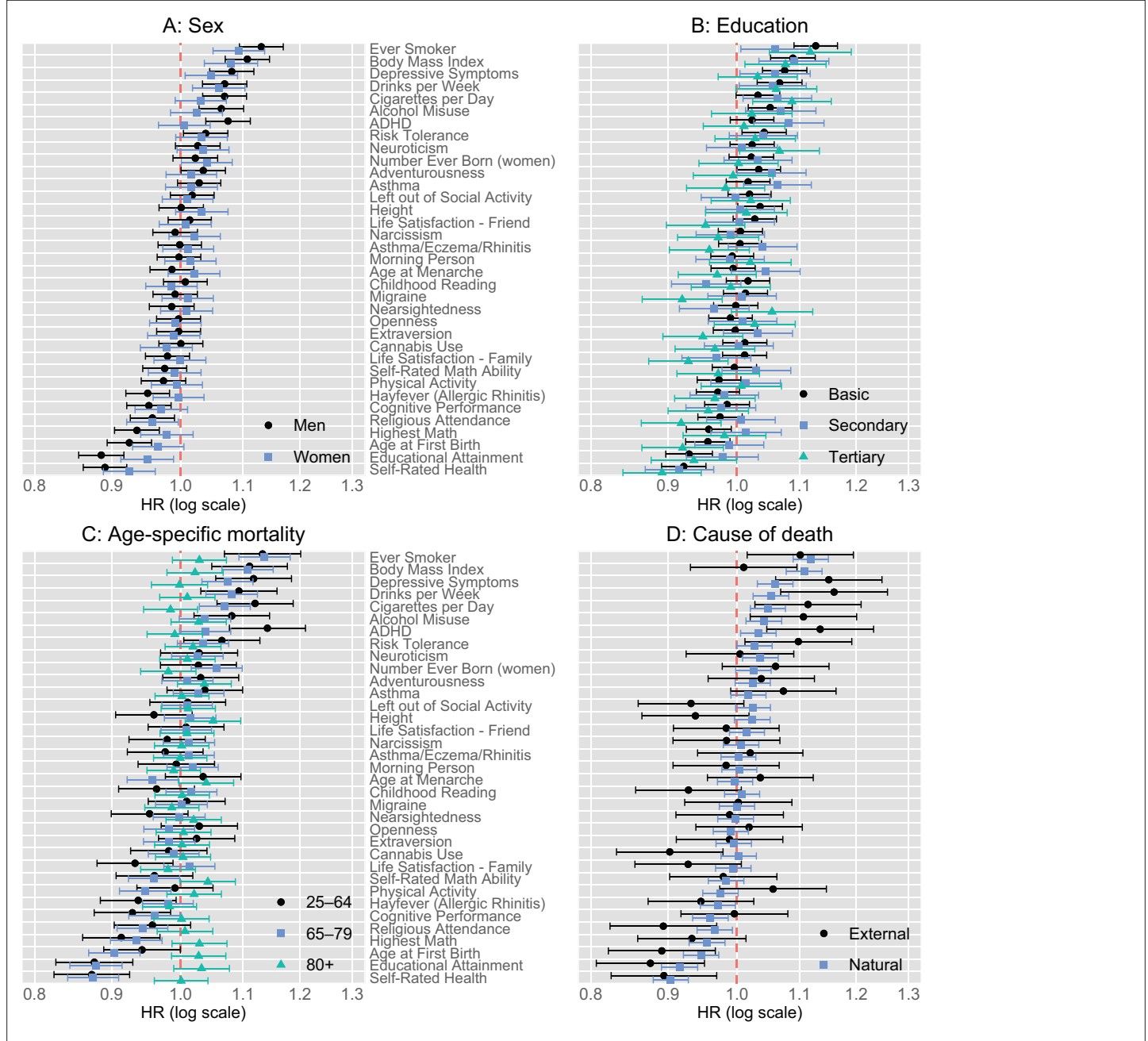

**Figure 2.** Hazard ratios of polygenic indices for all-cause mortality by sex (Panel **A**), educational group (Panel **B**), age-specific mortality follow-up period (Panel **C**), and cause of death (Panel **D**). Estimates from Cox proportional hazards models adjusted for indicators for the baseline year, sex and 10 first principal components of the genome. Capped bars are 95% confidence intervals. N=40,097 individuals. For a table of corresponding estimates, see **Supplementary file 1F**. Abbreviations: HR = Hazard ratio; ADHD = Attention deficit hyperactivity disorder.

for 80+ year olds, p=7*10⁻⁴ for difference, p=0.012 after multiple-testing adjustment) and depressive symptoms (HR = 1.12, 95% CI 1.06; 1.18 for 25–64 year-olds, HR = 1.00, 95% CI 0.96; 1.04 for 80+ year olds, p=0.002 for difference, p=0.032 after multiple-testing adjustment). Additionally, the difference in HRs between these age groups achieved significance after multiple-testing adjustment at the conventional 5% level for PGIs of cigarettes per day, educational attainment, and ever smoking.

Panel D displays that most PGIs had stronger associations with external (accidents, suicides, violence, and alcohol-related causes of death) than natural causes of death. An exception was the PGI of BMI that had a larger HR for natural (HR = 1.11, 95% CI 1.08; 1.14) than external causes of death (HR = 1.01, 95% CI 0.93; 1.10). The HR differences between external and natural causes of death

**Table 1.** Hazard ratios of selected polygenic indices and corresponding phenotypes for all-cause mortality (N: 37,548 individuals; 5407 deaths).

All presented variables were standardised to standard-deviation units. For corresponding models with categorical phenotypes, see *Supplementary file 1G*.

| | Model 1 a–1 l | | | Model 2 a–2 f | | | Model 3 a & 3b | | | Model 4 | | |
| | | 95% CI | | | 95% CI | | | 95% CI | | | 95% CI | |
| | HR | lower | upper | HR | lower | upper | HR | lower | upper | HR | lower | upper |
|---|---|---|---|---|---|---|---|---|---|---|---|---|
| Smoking | 1.41 | 1.37 | 1.45 | 1.39 | 1.35 | 1.44 | 1.35 | 1.31 | 1.40 | 1.34 | 1.29 | 1.38 |
| PGI-ever smoking | 1.12 | 1.09 | 1.15 | 1.07 | 1.04 | 1.10 | 1.07 | 1.04 | 1.10 | 1.04 | 1.01 | 1.08 |
| BMI | 1.07 | 1.04 | 1.10 | 1.05 | 1.01 | 1.08 | 1.02 | 0.99 | 1.05 | 1.00 | 0.97 | 1.04 |
| PGI-BMI | 1.10 | 1.07 | 1.13 | 1.08 | 1.05 | 1.11 | 1.05 | 1.02 | 1.09 | 1.03 | 1.00 | 1.06 |
| Depression indicators | 1.16 | 1.14 | 1.19 | 1.16 | 1.13 | 1.19 | 1.06 | 1.03 | 1.08 | 1.05 | 1.03 | 1.08 |
| PGI-depressive symptoms | 1.07 | 1.04 | 1.10 | 1.06 | 1.03 | 1.09 | 1.02 | 0.99 | 1.05 | 1.00 | 0.97 | 1.03 |
| Alcohol intake | 1.16 | 1.13 | 1.19 | 1.15 | 1.12 | 1.18 | 1.12 | 1.09 | 1.14 | 1.11 | 1.08 | 1.14 |
| PGI drinks per week | 1.06 | 1.03 | 1.09 | 1.05 | 1.02 | 1.08 | 1.05 | 1.02 | 1.08 | 1.03 | 1.00 | 1.06 |
| Education years | 0.86 | 0.83 | 0.88 | 0.87 | 0.84 | 0.90 | 0.90 | 0.88 | 0.93 | 0.91 | 0.88 | 0.94 |
| PGI educational attainment | 0.91 | 0.89 | 0.94 | 0.94 | 0.92 | 0.97 | 0.96 | 0.93 | 0.99 | 1.02 | 0.99 | 1.05 |
| Self-rated health | 0.74 | 0.72 | 0.76 | 0.75 | 0.73 | 0.77 | 0.77 | 0.75 | 0.80 | 0.78 | 0.75 | 0.80 |
| PGI self-rated health | 0.90 | 0.88 | 0.93 | 0.94 | 0.91 | 0.96 | 0.95 | 0.92 | 0.99 | 0.98 | 0.94 | 1.01 |

All models adjusted for baseline covariates: indicators of the baseline year, sex, and ten first principal components of the genome.
Models 1: Baseline covariates + each phenotype/PGI separately.
Models 2: PGI and corresponding phenotype mutually adjusted.
Model 3 a: Phenotypes mutually adjusted, Model 3b: PGIs mutually adjusted.
Model 4: Full model.
HR=Hazard ratio.
95% CI=95% confidence interval.
PGI=Polygenic index.
BMI=Body mass index.

were nominally significant at the conventional 5% level for cannabis use (p=0.016), drinks per week (p=0.028), left out of social activity (p=0.029), ADHD (p=0.031), BMI (p=0.035), and height (p=0.049), but none of these differences remained significant after adjusting for 35 multiple tests. For external causes, the strongest associations were observed for the PGI for drinks per week (HR = 1.16, 95% CI 1.07; 1.26), depressive symptoms (HR = 1.15, 95% CI 1.06; 1.25), educational attainment (HR = 0.88, 95% CI 0.81; 0.95), and ADHD (HR = 1.14, 95% CI 1.05; 1.23). Twelve PGIs had HRs ≥1.1 (drinks per week, depressive symptoms, cigarettes per day, ADHD, alcohol misuse, ever smoking, risk tolerance) or ≤0.9 (cannabis use, self-rated health, religious attendance, age at first birth, and educational attainment) per 1 SD difference in PGI, whereas only three PGIs had HRs ≥1.1 or≤0.9 for natural causes of death (ever smoking, BMI, and self-rated health). HRs of natural causes of death followed similar patterns as all-cause mortality, which was expected, as they constituted 90% of all observed deaths.

*Table 1* shows HRs of the six most predictive PGIs (ever smoking, BMI, depressive symptoms, drinks per week, educational attainment, and self-rated health) and their corresponding phenotypes (smoking, self-rated health, years of education, BMI, number of depression indicators, and alcohol intake per week) for all-cause mortality. Models 1 a–1 l present HRs of each variable adjusted only for baseline covariates. All phenotypes except BMI had stronger associations with mortality than their corresponding PGIs. Models 2 a–2 f adjust for each phenotype and its corresponding PGI simultaneously. HRs of phenotypes were only slightly attenuated, whereas for PGIs, this attenuation was around one-third on average. Nevertheless, each PGI was clearly associated with mortality even after adjusting for its phenotype, and all estimated 95% CIs excluded one in Models 2 a–2 f. Model 3 a adjusted for all six phenotypes and 3b for all six PGIs simultaneously. The most substantial attenuation

**Table 2.** Hazard ratios of selected polygenic indices for all-cause mortality regarding belonging to a risk category in the related phenotypes.

'Without phenotype' categories consist of: Never smokers (N: 21,189), BMI 18.5–24.9 (N:15,092), no depression indicators (N: 31,089), non-alcohol drinkers (N: 5498), higher-tertiary-degree holders (N: 4765), the best self-rated health (5-point scale, N: 7566). 'With phenotype' categories are those in the population sample with valid information on the phenotype in question excluding the 'without risk factor' category.

| | Without phenotype | | | With phenotype | | |
|---|---|---|---|---|---|---|
| | HR | 95% CI | | HR | 95% CI | |
| | | lower | upper | | lower | upper |
| PGI-ever smoking | 1.07 | 1.03 | 1.11 | 1.09 | 1.05 | 1.12 |
| PGI-BMI | 1.13 | 1.08 | 1.19 | 1.09 | 1.06 | 1.13 |
| PGI-depressive symptoms | 1.05 | 1.02 | 1.08 | 1.06 | 1.01 | 1.12 |
| PGI-drinks per week | 1.09 | 1.04 | 1.15 | 1.07 | 1.03 | 1.10 |
| PGI-educational attainment | 0.88 | 0.76 | 1.02 | 0.93 | 0.90 | 0.96 |
| PGI-self-rated health | 0.88 | 0.80 | 0.96 | 0.91 | 0.89 | 0.94 |

All models adjusted for indicators of the baseline year, sex, and ten first principal components of the genome.
HR=Hazard ratio.
95% CI=95% confidence interval.
PGI=Polygenic index.
BMI=Body mass index.

was observed for depression indicators and the PGI of depressive symptoms. Finally, Model 4 adjusted for all phenotypes and PGIs simultaneously. In Model 4, PGIs had modest independent associations, the strongest observed being for the PGI of smoking (HR = 1.04, 95% CI 1.01; 1.08), PGI of BMI (HR = 1.03, 95% CI 1.00; 1.06), and PGI of drinks per week (HR = 1.03, 95% CI 1.00; 1.06). *Supplementary file 1G* presents corresponding analyses with categorised phenotypes, indicating curvilinear mortality patterns for BMI and alcohol intake. Although HRs of these phenotypes in *Table 1* should thus be interpreted with caution, HRs of PGIs were consistent between the analyses presented in *Table 1* and *Supplementary file 1G*.

*Supplementary file 1H* presents information criteria and significance tests on corresponding models. Models with PGI +phenotype (Models 2 a–f) showed improvement over models with the phenotype only (Models 1 a, 1 c, 1e, 1 g, 1i, 1k, with a p=0.0006 or lower) in terms of both Akaike information criterion (AIC) as well as Bayesian (Schwarz) information criterion (BIC) with a p=0.0006 or lower in all comparisons. The full Model 4 again showed improvement over the model with all PGIs jointly (Model 3b, with a p=0.0002 or p=0.00002, depending on continuous/categorical phenotype measurement), which had a lower AIC but not BIC.

*Table 2* shows the associations between these six PGIs (ever smoking, BMI, depressive symptoms, drinks per week, educational attainment, and self-rated health) and all-cause mortality stratified by whether an individual was lacking the phenotype risk factor in question. We did not observe evidence for substantial differences in the PGI–mortality HRs between individuals with and without these risk factors in their corresponding phenotype. Furthermore, the only PGI that showed consistent attenuation in their HRs compared to the analysis on the whole sample (presented in *Table 1* and *Figure 1*) was the PGI of ever smoking (HR = 1.07, 95% CI 1.03; 1.11 for never smokers; HR = 1.09, 95% CI 1.05; 1.12 for others; compare HR = 1.12, 95% CI 1.09; 1.15 for unstratified population analysis in Model 1b of *Table 1*).

## Discussion

We investigated the association between 35 PGIs – mostly related to social, psychological, and behavioural traits, or typically non-fatal health conditions – and mortality using a population-representative register-linked sample from Finland with up to 25-year mortality follow-up. PGIs most

strongly associated with mortality were typically related to the best-established phenotypic mortality risk factors, including smoking, body mass index, depression, alcohol use, education and self-rated health (*Hummer and Lariscy, 2011*; *Himes, 2011*; *Jylhä, 2011*). Although the majority of the investigated PGIs had negligible associations with the risk of death, the strongest associations observed were about a 10% difference in the mortality hazard for 1 SD difference in PGI. Given the severity of the outcome, these associations cannot be disregarded as trivial, particularly when considering individuals with particularly high or low PGIs. Our within-sibship analyses showed broadly similar results and thus do not indicate that these PGI–mortality associations were systematically inflated due to population stratification or related biases. Limited previous literature exists overall on PGI–mortality associations, particularly for other than disease- or biomarker-related PGIs. Nevertheless, the associations of PGIs of smoking, alcohol consumption, depression and BMI that we observed for Finland were roughly comparable to or moderately stronger than what was observed in a previous study on the UK Biobank for PGIs unadjusted for each other (*Meisner et al., 2020*) and in US-based Health and Retirement Study for mutually adjusted PGIs (*Karlsson Linnér and Koellinger, 2022*).

The investigated PGIs were slightly more predictive of mortality among men than women across the board. This aligns with sex differences for all-cause mortality observed for many social-level mortality risk phenotypes such as socioeconomic position and marital status (*Case and Paxson, 2005*; *Koskinen and Martelin, 1994*; *Wang et al., 2020*), but similar excess risk among men is not consistently observed on more physiologically proximate or behavioural mortality risk phenotypes such as obesity, alcohol use, and smoking (*Gellert et al., 2012*; *Griswold et al., 2018*; *Hu et al., 2005*). We also evaluated but did not observe differences in PGI–mortality associations between education groups. The PGIs were more predictive of death at younger ages, whereas among those who survived to age 80, PGIs made hardly any difference in mortality risk. Such an age-related heterogenous pattern is consistent with a previous study analysing age–PGI associations on common diseases in the UK Biobank (*Jiang et al., 2021*). In contrast, a previous study found stronger HRs in older age groups when directly identifying SNPs associated with mortality (*Ganna et al., 2013*). A possible explanation for such an 'age as a leveller' pattern (see *Hoffmann et al., 2019*; *Hu et al., 2018*) may lie in the increasing importance of acute mortality-risk-enhancing factors towards the end of life, including emerging and progressing illness and biological ageing, which trump more distant and indirect mechanisms (*Hoffmann et al., 2019*; *Hoffmann, 2011*). Such age-specific heterogeneity also has methodological implications. Researchers analysing PGI–mortality associations using samples of (disproportionately) older individuals should be cautious in generalising their results to the overall population due to potential survivorship bias (*Digitale et al., 2023*; *Akimova et al., 2021*).

In general, PGIs were more strongly predictive of external than natural causes of death, and this was particularly evident for many psychological and behavioural PGIs, including alcohol drinks per week, ADHD, depressive symptoms, religious attendance, and risk tolerance. Only the PGI of BMI showed a clearly stronger association with natural causes. It is worth noting that despite alcohol-related deaths being included in external causes, smoking-related PGIs predicted both external and natural mortality in a roughly consistent manner, and the PGI of cannabis use even had a negative association with external mortality. This suggests that despite a substantial shared genetic aetiology between different addictions (*Liu et al., 2019*; *Prom-Wormley et al., 2017*), the genetic architecture between the use of different substances also differs importantly from the perspective of mortality risk.

Among those PGIs that were most predictive of mortality, their associations tended to be roughly one-third of the strength of the respective phenotypes when mutually adjusted, although with substantial variation between phenotypes. Additionally, PGIs were also predictive among those who lack the phenotypic risk factor. These two observations imply that PGIs provide additional information on the risk of death even when the phenotypic measures are available. The potential advantage of PGIs in research and healthcare relative to phenotype is that they need to be measured only once, whereas for many phenotypes the most precise monitoring would require extensive longitudinal measures. This is particularly relevant for phenotypes related to specific points of the life course, for example, on health-related factors typically manifesting at older age. This suggests that the independent association of PGIs might be stronger if we had even longer mortality follow-up. Additionally, PGIs capture liabilities on a continuum, which may offer an advantage in risk assessment compared to phenotypes that are typically measured through binary diagnoses or with limited categories. PGIs also avoid some

potential forms of measurement error, such as those related to self-reporting or short-term variations over time.

The strengths of this study included population-representative data with high response rates linked to a long register-based follow-up with minimal attrition-related biases. On the other hand, PGIs have known limitations since they incompletely capture the genetic liability, are agnostic regarding the specific biological mechanisms and may also capture environmental signals (*Burt, 2024*). Within-sibship analysis allowed us to evaluate and mitigate population stratification and other related biases in the analyses; however, at the cost of power and increased sensitivity to measurement error (*Gustavson et al., 2024*). Overall, these analyses confirmed our main findings. In addition, comparability of the PGIs may be affected by differences in the GWASs that underlie them, for example in their sample sizes and phenotype measurement quality. Finally, despite the Schoenfeld residual correlations suggesting some violation of the proportional hazards assumption, such correlations were present for phenotypes (instead of PGIs) that were not of the primary interest in the analysis.

To conclude, PGIs related to the best-established phenotypic risk factors had the strongest associations with mortality. Particularly for deaths occurring at a younger age, PGIs confer additional information on mortality risk, even when information on the related phenotype is available, and within-sibship analysis did not suggest that such associations were systematically inflated due to population phenomena.

## Acknowledgements

We thank Aysu Okbay for the weighting of the PGIs used. We would like to thank the research participants and employees of 23andMe, Inc for making this work possible. The genetic samples used for the research were obtained from the THL Biobank (study number: THLBB2020_8/THLBB2023_51), and we thank all study participants for their generous participation in the THL Biobank.

## Additional information

### Funding

| Funder | Grant reference number | Author |
|---|---|---|
| European Research Council | 101019329 | Pekka Martikainen |
| Research Council of Finland | 352543-352572 | Pekka Martikainen |
| Jane ja Aatos Erkon Säätiö | 210046 | Pekka Martikainen |
| the Max Planck Society | 5714240218 | Pekka Martikainen |
| Helsingin Yliopisto | 77204227 | Pekka Martikainen |
| Cities of Helsinki, Vantaa and Espoo | 4706914 | Pekka Martikainen |
| Research Council of Finland | 350399 | Stefano Lombardi |
| Research Council of Finland | 345219 | Pekka Martikainen |
| Research Council of Finland | 136528219 | Pekka Martikainen |
| Research Council of Finland | 136528212 | Pekka Martikainen |

The funders had no role in study design, data collection and interpretation, or the decision to submit the work for publication.

## Author contributions
Hannu Lahtinen, Conceptualization, Data curation, Formal analysis, Funding acquisition, Investigation, Visualization, Methodology, Writing – original draft; Jaakko Kaprio, Resources, Data curation, Methodology, Writing - review and editing; Andrea Ganna, Resources, Data curation, Writing - review and editing; Kaarina Korhonen, Data curation, Methodology, Project administration, Writing - review and editing; Stefano Lombardi, Methodology, Writing - review and editing; Karri Silventoinen, Writing - review and editing; Pekka Martikainen, Data curation, Supervision, Funding acquisition, Project administration, Writing - review and editing

## Author ORCIDs
Hannu Lahtinen ⬛ https://orcid.org/0000-0003-0910-823X
Jaakko Kaprio ⬛ https://orcid.org/0000-0002-3716-2455
Kaarina Korhonen ⬛ https://orcid.org/0000-0001-8499-2008

## Ethics
The Finnish Social and Health Data Permit Authority (Findata) has accepted the use of clinical data (THL/4725/14.02.00/2020; THL/1423/14.06.00/2022), and the THL Biobank has approved the use of genetic data (THLBB2020_8; THLBB2023_51) and the data linkage to the Finnish population registers (TK-53-876-20; TK/2041/07.03.00/2023). All participants gave their informed consent.

Reviewer #1 (Public review): https://doi.org/10.7554/eLife.107496.3.sa1
Reviewer #2 (Public review): https://doi.org/10.7554/eLife.107496.3.sa2
Author response https://doi.org/10.7554/eLife.107496.3.sa3

## Additional files

### Supplementary files
Supplementary file 1. Supplementary Tables.

MDAR checklist

### Data availability
The code used for data preparation and analysis can be found in https://github.com/halahti/eLife26/ (copy archived at *Lahtinen, 2026*). The statistics used in *Figures 1 and 2* can be found in *Supplementary file 1E and F*. Due to data protection regulations of the biobank and national register-holders, the data are confidential, and we are not allowed to make the individual-level data available to third parties, even as anonymized. Datasets are available from the THL Biobank on written application and following the instructions given on the website of the Biobank (https://thl.fi/en/research-and-development/thl-biobank/for-researchers, contact by email: admin.biobank (at) thl.fi). Register linkage from these data may be applied for from Findata (https://findata.fi/en/permits/, contact by email: info( at)findata.fi.) and Statistics Finland (http://www.stat.fi/tup/mikroaineistot/index_en.html, contact by email: tutkijapalvelut(at)stat.fi). The use of those PGIs that include samples from 23andme data collections requires their own permission to use by the 23andMe, Inc (apply.research (at) 23andme.com).

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
