## [Editor Report · eLife Assessment]

This **important** study reports **convincing** evidence of associations between 35 polygenic indices (PGIs) for social, behavioural, and psychological traits, as well as other health conditions (e.g., BMI) and all-cause mortality, based on data from Finnish population-based surveys and a twin cohort linked to administrative registers. PGIs for education, depression, alcohol use, smoking, BMI, and self-rated health showed the strongest associations with all-cause mortality, in the order of ~10% increment in risk per PGI standard deviation. Effect sizes from twin-difference analyses tended to be slightly larger than those from population cohorts, a pattern opposite that generally observed when testing PGI associations with their target phenotypes, and supporting the robustness of findings to confounding by population stratification.

---

## [Referee Report · Reviewer #1 (Public review)]

Lahtinen et al. evaluated the association between polygenic scores and mortality. This question has been intensely studied (Sakaue 2020 Nature Medicine, Jukarainen 2022 Nature Medicine, Argentieri 2025 Nature Medicine), where most studies use PRS as an instrument to attribute death to different causes. The presented study focuses on polygenic scores of non-fatal outcomes and separates the cause of death into "external" and "internal". The majority of the results are descriptive, and the data doesn't have the power to distinguish effect sizes of the interesting comparisons: (1) differences between external vs. internal (2) differences between PGI effect and measured phenotype.

Comments on revised version:

The authors answered my concerns well. I don't have any further comments.

---

## [Referee Report · Reviewer #2 (Public review)]

Summary:

This study provides a comprehensive evaluation of the association between polygenic indices (PGIs) for 35 lifestyle and behavioral traits and all-cause mortality, using data from Finnish population- and family-based cohorts. The analysis was stratified by sex, cause of death (natural vs. external), age at death, and participants' educational attainment. Additional analyses focused on the six most predictive PGIs, examining their independent associations after mutual adjustment and adjustment for corresponding directly measured baseline risk factors.

Strengths:

Large sample size with long-term follow-up.

Use of both population- and family-based analytical approaches to evaluate associations.

Comments on revised version:

I am happy with the revision. No further comments.

---

## [Author Response]

The following is the authors’ response to the original reviews.

**Public Reviews:**

**Reviewer #1 (Public review):**
Lahtinen et al. evaluated the association between polygenic scores and mortality. This question has been intensely studied (Sakaue 2020 Nature Medicine, Jukarainen 2022 Nature Medicine, Argentieri 2025 Nature Medicine), where most studies use PRS as an instrument to attribute death to different causes. The presented study focuses on polygenic scores of non-fatal outcomes and separates the cause of death into "external" and "internal". The majority of the results are descriptive, and the data doesn't have the power to distinguish effect sizes of the interesting comparisons: (1) differences between external vs. internal (2) differences between PGI effect and measured phenotype. I have two main comments:(1) The authors should clarify whether the p-value reported in the text will remain significant after multiple testing adjustment. Some of the large effects might be significant; for example, Figure 2C

We have now added Benjamini-Hochberg multiple-testing adjusted p-values in the text each time we present nominal p-values. Additionally, supplementary tables S5 and S6 provide multiple-adjusted p-values for all analysed PGIs.

Although this was not always the case, many comparisons remained significant after multiple testing adjustments, especially in Figure 2C that the reviewer commented on. In the revised version, we have placed more emphasis on describing these HRs that have low p-values after multiple-test adjustment. The revised text for Figure 2C in the Results section now reads:

Panel C analyses mortality in three age-specific follow-up periods. The PGIs were more predictive of death in younger age groups, although the difference between the 25–64 and 65–79 age groups was small, except for the PGI of ADHD (HR=1.14, 95% CI 1.08; 1.21 for 25–64-year-olds; HR=1.04, 95% CI 1.00; 1.08 for 65–79-year-olds; p=0.008 for difference, p=0.27 after multiple-testing adjustment). PGIs predicted death only negligibly among those aged 80+, and the largest differences between the age groups 25–64 and 80+ were for PGIs of self-rated health (HR 0.87, 95% CI 0.82; 0.93 for 25–64-year-olds, HR 1.00, 95% CI 0.94; 1.04 for 80+ year-olds, p=2*10^-4^ for difference, p=0.006 after multiple-testing adjustment), ADHD (HR 1.14, 95% CI 1.08; 1.21 for 25–64-year-olds, HR 0.99, 95% CI 0.95; 1.03 for 80+ year-olds, p=7*10^-4^ for difference, p=0.012 after multiple-testing adjustment) and depressive symptoms (HR 1.12, 95% CI 1.06; 1.18 for 25–64-year-olds, HR 1.00, 95% CI 0.96; 1.04 for 80+ year-olds, p=0.002 for difference, p=0.032 after multiple-testing adjustment). Additionally, the difference in HRs between these age groups achieved significance after multiple testing adjustment at the conventional 5% level for PGIs of cigarettes per day, educational attainment, and ever smoking.

We have also included the recent study by Argentieri et al. (2025) in the literature review, which was missing from our previous version. We appreciate the reference. Other references mentioned were already included in the previous version of the manuscript.

(note that the small prediction accuracy of PGI in older age groups has been extensively studied, see Jiang, Holmes, and McVean, 2021, PLoS Genetics).

We would like to thank the reviewer for suggesting the relevant reference by Jiang et al. We have now expanded on the discussion of age-specific differences in the discussion section and included this reference.

(2) The authors might check if PGI+Phenotype has improved performance over Phenotype only. This is similar to Model 2 in Table 1, but slightly different.

The reviewer raises an interesting angle to approach the analysis. We have now added an analysis assessing the information criteria and the significance of improvement between nested models in Supplementary table S8. All the tested PGI+phenotype models show improvement over the phenotype-only model that is statistically significant at all conventional levels when tested by likelihood-ratio tests between nested models . Additionally, improvement was found when using Akaike and Bayesian (Schwarz) information criteria (albeit sometimes modest in size). We have added a passage in the results section briefly summarising this analysis:

Supplementary table S8 presents information criteria and significance tests on corresponding models. Models with PGI+phenotype (Models 2a–f) showed improvement over models with the phenotype only (Models 1a, 1c, 1e, 1g, 1i, 1k, with a p=0.0006 or lower) in terms of both Akaike information criterion (AIC) as well as Bayesian (Schwarz) information criterion (BIC) with a p=0.0006 or lower in all comparisons. The full Model 4 again showed improvement over the model with all PGIs jointly (Model 3b, with a p=0.0002 or p=0.00002, depending on continuous/categorical phenotype measurement), which had a lower AIC but not BIC.

**Reviewer #2 (Public review):**
Summary:This study provides a comprehensive evaluation of the association between polygenic indices (PGIs) for 35 lifestyle and behavioral traits and all-cause mortality, using data from Finnish population- and family-based cohorts. The analysis was stratified by sex, cause of death (natural vs. external), age at death, and participants' educational attainment. Additional analyses focused on the six most predictive PGIs, examining their independent associations after mutual adjustment and adjustment for corresponding directly measured baseline risk factors.Strengths:Large sample size with long-term follow-up.Use of both population- and family-based analytical approaches to evaluate associations.Weaknesses:It is unclear whether the PGIs used for each trait represent the most current or optimal versions based on the latest GWAS data.

To our reading, this comment is closely related to the “recommendations for the author” number 3 by reviewer 2, and we thus address them together.

If the Finnish data used in this study also contributed to the development of some of the PGIs, there is a risk of overestimating their associations with mortality due to overfitting or "double-dipping." Similar inflation of effect sizes has been observed in studies using the UK Biobank, which is widely used for PGI construction.

To our reading, this comment is closely related to the “recommendations for the author” 4 by reviewer 2, and we thus address them together.

**Recommendations for the authors:**

**Reviewer #1 (Recommendations for the authors):**
Specific comments:(1) Cited reference 1 also investigated the PRS association with life span; cited reference 8 explains PRS association with healthy lifespan. Can authors be clearer about what is new in the context of these references? Specifically, what are the PGIs studied here that were not analyzed in the cited analyses?

Although some previous studies on the topic do exist, our analysis arguably has novelty in touching upon several unstudied or scarcely studied themes. These include:

A set of PGIs focusing on social, psychological, and behavioural phenotypes or PGIs for typically non-fatal health conditions.

An assessment of direct genetic effects/ confounding with a within-sibship design.

An assessment of potential heterogeneous effects by several socio-demographic characteristics.

An analysis of external causes of deaths (which can be hypothesised to be particularly relevant here, given the choice of our PGIs not focusing directly on typical causes of death).

A detailed assessment of the interplay of the most predictive PGIs with their corresponding phenotypes.

We have substantially revised the Introduction section focusing on making these novel contributions more explicit.

(2) In the Methods section, it is not very clear why the authors specifically study the "within-sibship" samples. Is this for avoiding nurturing effects from parental genotypes or for controlling assortative mating? The authors should clarify the rationale behind the design.

The substance-related rationale behind this approach was briefly discussed in the Introduction section while in the Methods section, we focused more on the technical description of our analyses. However, it is certainly worthwhile to clarify to the reader why within-sibship methods have been used. The revised passage in the methods section now states:

“In addition to this population sample, we used a within-sibship analysis sample to assess the extent of direct and indirect genetic associations captured by the PGIs, as discussed in the introduction.”

(3) Residual correlations of PGIs were no more than 0.050..." As a minor comment, since PGIs is a noisy variable, the correlation would be low; however, I don't think there are better ways to evaluate Cox assumptions, and in many cases, this assumption is not correct for strong predictors.

Yes, these points are true. Overall, it is often implausible that empirical distributions exactly match distributional assumptions in statistical models. For example, it may not be realistic to expect that the mortality hazards across categories of independent variables stay exactly proportional during long mortality-follow-ups; some deviations from constant proportions are almost inevitable. However, there are reasonable grounds to argue that in case of moderate violations of the proportional hazards assumption, the estimates still remain interpretable for practical uses. They can be read as approximating average relative hazards over the study period (for discussion, see pages 42–47 in Allison P. 2014. Event history and survival analysis: Regression for longitudinal event data (second edition). Thousand Oaks: SAGE).

(4) "PGI of ADHD (HR=1.08 95%CI 1.04;1.11 among men; HR=1.01 95%CI 0.97;1.05 among women; p=0.012 for difference)." Is this difference significant after multiple testing correction?

We have presented multiple-testing adjusted p-values together with nominal ones in this and in all other instances where they are mentioned in the text. Additionally, Supplementary tables S5–S6 present multiple-adjusted p-values for each PGIs studied.

(5) "Panel D displays that most PGIs had stronger associations with external (accidents, violent, suicide, and alcohol related deaths) than natural causes of death." Similar to the comment above, are there any results that are significantly different between internal and external?

We have added the p-values of those variables that had larger differences in the revised text. Quoting from the revised article: “The HR differences between external and natural causes of death were nominally significant at the conventional 5% level for cannabis use (p=0.016), drinks per week (p=0.028), left out of social activity (p=0.029), ADHD (p=0.031), BMI (p=0.035) and height (p=0.049), but none of these differences remained significant after adjusting for 35 multiple tests. “

(6) Table 1: The effect of the phenotype is stronger than the PGI; this is expected as PGI is a weak predictor and can be considered as "noised" measurement of true genetic value (Becker 2021 Nature Human behavior). Is there a way to adjust for the impact of noise in PGI at tagging genetic value and compare if the PGI effect is different from the phenotype effect?

PGIs are certainly imperfect measures that contain a lot of noise. However, extracting new information from what is unknown is an extremely demanding exercise, and still further complicated for example, by that we do not know the exact benchmark of total genetic effect we should be aiming at. Different methods of heritability estimation, for instance, often give dramatically differing results – for reasons that are still up to scrutiny.

We are thus not familiar with a method that could achieve satisfactory answer for this challenging task.

**Reviewer #2 (Recommendations for the authors):**
(3) Justification and Selection of PGIs:For several traits, such as BMI, multiple polygenic indices (PGIs) are currently available. The criteria used to select specific PGIs for this study are not clearly described. A more systematic and reproducible approach-for example, leveraging metadata from the PGS Catalog-could strengthen the justification for PGI selection and enhance the study's generalizability.

There are numerous PGIs developed in the extensive GWAS literature, but a finite set of PGIs always needs to be chosen for any analysis. The rationale behind our decision to include every PGI from the repository of Becker et al. 2021 (full reference in the manuscript, see also https://www.thessgac.org/pgi-repository) that was available for the Finnish data (including the possibility to exclude overlapping samples, see our response to the next comment for more discussion) was to provide rigorous analysis by limiting the researchers degrees of freedom in arbitrarily choosing PGIs. Although it would have been tempting to not use some PGIs that were not expected to substantially correlate with mortality, we believe that our conservative strategy increases the credibility of the reported p-values, particularly the multiple adjustment should now work as intended.

We also mention now this rationale when discussing the chosen PGIs in the methods section: “As the independent variables of main interest, we used 35 different PGIs in the Polygenic Index repository by Becker et al., which were mainly based on GWASes using UK Biobank and 23andMe, Inc data samples, but also other data collections. They were tailored for the Finnish data, i.e., excluding overlapping individuals between the original GWAS and our analysis and performing linkage-disequilibrium adjustment. We used every single-trait PGI defined in the repository (except for subjective well-being, for which we were unable to obtain a meta-analysis version that excluded the overlapping samples). By limiting the researchers’ freedom in selecting the measures, this conservative strategy should increase the validity of our estimates, particularly with regards to multiple-testing adjusted p-values.”

(4) Overlap Between PGI Training Data and Study Sample:The authors should describe any overlap between the data used to develop the PGIs and the current study sample. If such overlap exists, it may lead to overestimation of effect sizes due to "double-dipping." A discussion of this issue and its potential implications is warranted, as similar concerns have been raised in studies using UK Biobank data.

This is, fortunately, not a concern of our analysis. Overlapping samples were excluded in creating the PGIs that we used. We have now described this more clearly in the revised methods section.

(1) Clarify the Methodology for Family-Based Cox Analysis:It is unclear what specific method was used to perform Cox regression in the family-based analysis. Please provide additional methodological details. ”

We have described the method further and added an additional reference in the revision. The text now stands:

“We compared these models to the corresponding within-sibship models, using the sibship identifier as the strata variable. This method employs a sibship-specific (instead of a whole-sample-wide baseline hazard in the population models) baseline hazard, and corresponds to a fixed-effects model in some other regression frameworks (e.g., linear model with sibship-specific intercepts)”

(2) Clarify Timing of Measured Risk Factors Relative to Follow-Up:The main text should provide more detailed information regarding the timing of data collection for directly measured risk factors. Specifically, it should be clarified whether the measurements used correspond to the first available data for each individual after the start of follow-up, or if a different criterion was applied.

BMI, self-rated health, alcohol consumption and smoking status were measured at the baseline survey of each dataset. Education was registered as the highest completed degree up to the end of 2019. Depression was a composite of survey self-report (at the time of the baseline survey), as well as depression-related medicine purchases and hospitalizations over a two-year period before the start of the individual’s follow-up.

We have added more comprehensive information on the measurement of the phenotypes of interest in Supplementary table 2, including the timing of the measurement.